

# External kinetics of the kettlebell snatch in amateur lifters

James A. Ross[1], Justin W.L. Keogh[2,3,4], Cameron J. Wilson[1] and Christian Lorenzen[1]

[1] School of Exercise Science, Australian Catholic University, Melbourne, VIC, Australia
[2] Faculty of Health Sciences and Medicine, Bond University, Gold Coast, QLD, Australia
[3] Sports Performance Research Institute New Zealand, Auckland University of Technology, Auckland, New Zealand
[4] Cluster for Health Improvement, Faculty of Science, Health, Education and Engineering, University of the Sunshine Coast, Sunshine Coast, QLD, Australia

Corresponding author
James A. Ross,
james.ross33@yahoo.com

## ABSTRACT

**Background:** Kettlebell lifting has gained increased popularity as both a form of resistance training and as a sport, despite the paucity of literature validating its use as a training tool. Kettlebell sport requires participants to complete the kettlebell snatch continuously over prolonged periods of time. Kettlebell sport and weightlifting involve similar exercises, however, their traditional uses suggest they are better suited to training different fitness qualities. This study examined the three-dimensional ground reaction force (GRF) and force applied to the kettlebell over a 6 min kettlebell snatch set in 12 kettlebell-trained males.

**Methods:** During this set, VICON was used to record the kettlebell trajectory with nine infrared cameras while the GRF of each leg was recorded with a separate AMTI force plate. Over the course of the set, an average of $13.9 \pm 3.3$ repetitions per minute were performed with a 24 kg kettlebell. Significance was evaluated with a two-way ANOVA and paired $t$-tests, whilst Cohen's F (ESF) and Cohen's D (ESD) were used to determine the magnitude.

**Results:** The applied force at the point of maximum acceleration was $814 \pm 75$ N and $885 \pm 86$ N for the downwards and upwards phases, respectively. The absolute peak resultant bilateral GRF was $1,746 \pm 217$ N and $1,768 \pm 242$ N for the downwards and upwards phases, respectively. Bilateral GRF of the first and last 14 repetitions was found to be similar, however there was a significant difference in the peak applied force ($F_{(1.11)} = 7.42$, $p = 0.02$, ESF = 0.45). Unilateral GRF was found have a significant difference for the absolute anterior–posterior ($F_{(1.11)} = 885.15$, $p < 0.0001$, ESF = 7) and medio-lateral force vectors ($F_{(1.11)} = 5.31$, $p = 0.042$, ESF = 0.67).

**Discussion:** Over the course of a single repetition there were significant differences in the GRF and applied force at multiple points of the kettlebells trajectory. The kettlebell snatch loads each leg differently throughout a repetition and performing the kettlebell snatch for 6 min will result in a reduction in peak applied force.

## INTRODUCTION

Kettlebell sport, also referred to as girevoy sport (GS), competition originated in Eastern Europe in 1948 (*Tikhonov, Suhovey & Leonov, 2009*). In recent years, kettlebell lifting has gained increased popularity as both a form of resistance training and a sport. The kettlebell snatch is one of the most popular exercises performed with a kettlebell. The movement is an extension of the kettlebell swing, and involves swinging the kettlebell upwards from between the legs until it reaches the overhead position. To date, the barbell snatch has received much attention and reviews of the literature have demonstrated it to be an effective exercise for strength and power development (*Escamilla, Lander & Garhammer, 2000*; *Garhammer, 1993*). In contrast, the kettlebell snatch has only just started to receive research attention (*Falatic et al., 2015*; *Lake, Hetzler & Lauder, 2014*; *McGill & Marshall, 2012*; *Ross et al., 2015*).

In a classic kettlebell competition, the winner is the person who completes the most snatch lifts within a 10 min period. Current rules stipulate that the athlete can only change the hand holding the kettlebell once during this 10 min period. Additionally, to perform a valid repetition the kettlebell must be locked out motionless overhead at the conclusion of each repetition. The overhead position is known as fixation, which was found to have the lowest movement variability compared to the end of the back swing, and the midpoints of the upwards and downwards phases within its trajectory (*Ross et al., 2015*). It has been proposed that due to the kettlebell's unique shape and its resulting trajectory, the unilateral kettlebell snatch may be better suited for performing multiple repetitions than a single maximum effort (*Ross et al., 2015*). Specifically, the kettlebell snatch trajectory follows a 'C'-shaped path as it can move in between the athlete's legs (*Ross et al., 2015*), in contrast to an 'S'-shaped trajectory of the barbell snatch (*Ho et al., 2014*; *Newton, 2002*), which moves in front of the knees facilitating a double knee bend. In elite kettlebell sport, the kettlebell snatch also involves a downwards phase which follows a smaller radius compared to the kettlebell's upwards phase (*Ross et al., 2015*). The downwards phase gives the kettlebell snatch more of a cyclical natural than the barbell snatch, where the barbell is dropped from the overhead recovery position, thus providing a training stimulus in both the upwards and downwards phases.

The kettlebell snatch and barbell snatch move through a number of different phases that share some similarities. From the starting position, the barbell snatch has the following phases: first pull, transition, second pull and the catch phase (*Haff & Triplett, 2015*; *Ho et al., 2014*). In contrast, the kettlebell snatch starts at fixation and has the following phases: drop, re-gripping, back swing, forward swing, acceleration pull and hand insertion phase (*Ross et al., 2015*; *Rudnev, 2010*). The second pull has been shown to be the most powerful motion during the barbell snatch (*Garhammer, 1993*). Similarly, the acceleration pull phase has been suggested to be the most explosive phase of the kettlebell snatch (*Rudnev, 2010*).

There is currently little research on the kinetics of the kettlebell snatch. The only study to date recorded the bilateral ground reaction force (GRF) of the kettlebell swing

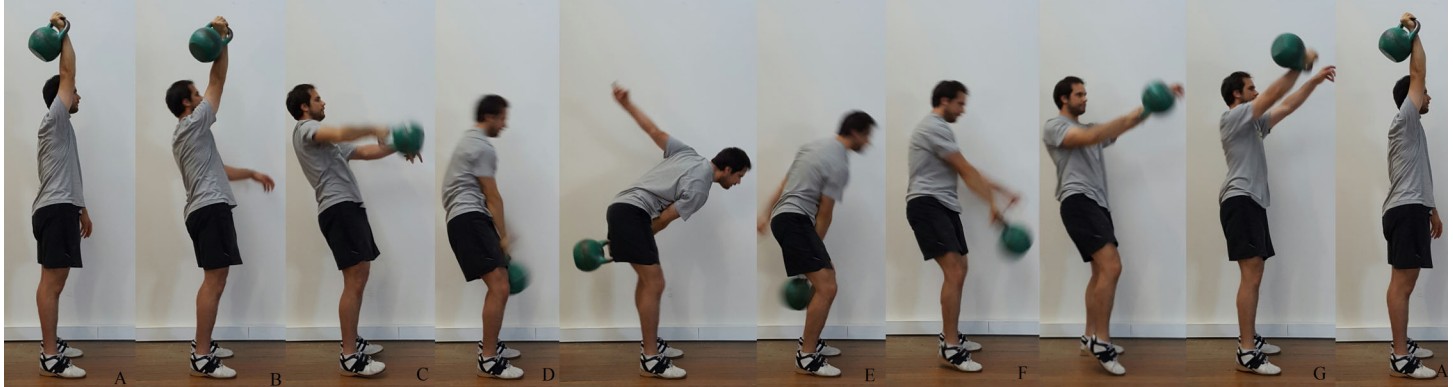

**Figure 1 Illustrates the phases of the kettlebell snatch.** (A) fixation, (B) drop, (C) re-gripping, (D) back swing, (E) forward swing, (F) acceleration pull, (G) hand insertion, (A) fixation.

and snatch (*Lake, Hetzler & Lauder, 2014*). The kettlebell snatch and two-handed swing were analysed over three sets of eight repetitions performed with the intention of achieving the maximum velocity possible, with horizontal and vertical work, impulse, mean force and power of the kettlebell snatch and swing calculated (*Lake, Hetzler & Lauder, 2014*). Both exercises had greater vertical impulse, work and mean force power than the horizontal equivalent regardless of phase (*Lake, Hetzler & Lauder, 2014*). The vertical component of the kettlebell snatch and two-handed swing were comparable, whilst the two-handed swing had a larger amount of work and rate of work performed in the horizontal plane (*Lake, Hetzler & Lauder, 2014*). One of the limitations was that GRF was investigated bilaterally when the movement is unilateral and is therefore likely to load the ipsilateral and contralateral legs differently (*Lauder & Lake, 2008*). This study investigated the hardstyle kettlebell snatch, which may have a different hip action, when contrasted with the kettlebell sport snatch. The hardstyle kettlebell snatch involves a single explosive leg extension. The upwards phase is described as being a swing, high pull and punch up (*Tsatsouline, 2006*). In contrast, the kettlebell sport snatch typically involves a double knee bend in an effort to improve the efficiency of the exercise. The double knee bend allows the kettlebell to transition forwards from the end of the back swing, before the explosive leg extension takes place (*Rudnev, 2010*). Although, there is inter-lifter variation within kettlebell sport technique (*Ross et al., 2015*), commonly kettlebell sport involves plantarflexion of the ipsilateral ankle during the explosive leg action (see Fig. 1), whilst this is not the case within the hardstyle snatch.

This study aims to build on the work by *Lake, Hetzler & Lauder (2014)* by investigating the unilateral GRF of the kettlebell snatch, throughout key positions of a single repetition and a prolonged set. In addition, force applied to the kettlebell by the lifter was also examined and will further the understanding of the kinetics of the key points of the trajectory outlined previously (*Ross et al., 2015*). These data will offer coaches an insight into the kinetic demands that the kettlebell snatch places upon the body providing insight to guide kettlebell exercise prescription.

## METHODS

### Study design

Twelve amateur kettlebell sport lifters performed 6 min of the kettlebell snatch exercise with one hand change, as is commonly performed in training by GS competitors. GRF was recorded with two AMTI force plates and kettlebell trajectory was simultaneously recorded with a nine-camera VICON motion analysis system. The GRF from the force plates allowed us to determine the external mechanical demands applied to the lifter and kettlebell system centre of mass, whilst the reverse kinematics calculated the force applied to the kettlebell. Force was determined using the kettlebell's known mass (kg) and the acceleration (m s$^{-2}$) determined via reverse kinematics. The aim was to identify the external demands placed upon each leg and the changes in kinetics during a prolonged kettlebell snatch set over 6 min. The dependent variables were resultant kettlebell force (N), resultant absolute and relative GRF (N) for: resultant, anterior–posterior, medio-lateral and vertical bilateral, GRF impulse (N s) and resultant velocity of the kettlebell (m s$^{-1}$). These were measured at the following time points: time of peak GRF, point of maximum kettlebell acceleration, point of maximum kettlebell velocity, end of back swing, lowest kettlebell point, midpoint and highest kettlebell point.

### Subjects

Twelve males with a minimum of three years kettlebell training experience (age 34.9 ± 6.6 years, height 182 ± 8.0 cm and mass 87.7 ± 11.6 kg, handgrip strength non-dominant 54.5 ± 8.0 kg and dominant 59.6 ± 5.5 kg) gave informed consent to participate in this study. They were free from injury and their training regularly included 6 min kettlebell snatch sets. Prior to taking part in the study, the participants performed 6.0 ± 2.1 training sessions per week, of which 3.3 ± 1.9 were with kettlebells. All had previously competed in kettlebell sport and kettlebell sport was the primary sport for nine of the 12 participants. A 24 kg kettlebell was selected, as this is the weight used by 'amateur' lifters within a kettlebell sport competition. This is in contrast to 32 kg weight for 'professional' lifters and 16 kg for 'novice' lifters. The Australian Catholic University's ethics review panel granted approval for this study to take place (ethics number 2012 21V). All participants gave written consent to take part in this research.

### Procedures

During a single testing session, athletes performed one 6 min kettlebell snatch set with a hand change taking place at the 3 min mark. A 6 min set was chosen as opposed to the GS standard 10 min set, as it was attainable for all subjects and is a common training set duration for non-elite kettlebell sport athletes. Handgrip strength was tested with a grip dynamometer with a standardised procedure 10 min pre-set and immediately post-test (*ACSM, 2013*). They were provided with chalk and sand paper (as this is standard competition practice) and asked to prepare the kettlebell as they would before training or competition. A range of professional-grade kettlebells of varying masses (Iron Edge, Australia) were available for the lifters to perform their typical warm ups. Following the athletes warm up, each 6 min set was performed with a professional-grade 24 kg kettlebell,

as is the standard for kettlebell sport within Australia. Three markers were used, one (26.6 mm × 25 mm) was placed on the front plate of the kettlebell, and two markers (14 mm × 12.5 mm in diameter) were placed on the kettlebell at the base of each side of the handle. The markers were placed in these positions to help avoid contact with the lifter during the set. Nine VICON infrared cameras (six MX 13+ and three T20-S) sampling at 250 Hz, were placed around two adjacent OR6 AMTI force plates sampling at 1,000 Hz. The point of origin was set in the middle of the platform, to calibrate the cameras' positions. The athlete was instructed to stand still with one foot on each plate and the kettlebell approximately 20 cm in front of him before the start of the 6 min set in order to process a static model calibration. A self-paced set was then performed as if they were being judged in a competition. To initiate the set, the kettlebell was pulled back between the legs.

VICON Nexus software was used to manually label markers, and a frame-by-frame review of each trial was performed to minimise error. Average marker position was computed at rest from initial position. The initial position of the markers was used to compute vectors from centroid to the centre of gravity. Kettlebell motion was computed using singular value decomposition of the marker transformations into a translation, a rotation and an error value (*Duarte, 2014*). Root mean square error was calculated and time steps with high error values were dropped from analysis. The centre of gravity locations were computed from the translation and rotation of the kettlebell geometry. A third order B-spline was used to interpolate and filter the three-dimensional trajectories using the python function ('`scipy.interpolate.splprep`'). The spline functions ('`knots`') were then used to compute the velocity and acceleration.

Time steps of the kettlebells trajectory that contained the kettlebell maximum velocity, maximum acceleration (peak resultant kettlebell force) and the following points: end of the back swing, lowest point, midpoints and highest point (overhead lockout position) were identified. At these time steps the resultant kettlebell force, resultant bilateral GRF, and resultant velocity were recorded. Time steps moving from the overhead lockout position to the end of the back swing were allocated a relative negative time in seconds, with the end of the back swing as zero. The time steps from the end of the back swing moving to the overhead lockout were given a positive relative time. Over the entire set at the point that peak bilateral absolute resultant force or peak resultant force for the ipsilateral and contralateral leg was reached, the three-dimensional force was reported. In addition to the entire set, the three-dimensional bilateral forces were reported for the first and last 14 repetitions. Fourteen repetitions were chosen because it was the closest whole number to the mean repetitions per minute performed by the subjects over the 6 min. The forces were presented in both absolute units and relative to each subject's body mass. As the majority of the work occurred between the end of the back swing and the midpoint of the upwards and downwards phases of its trajectory, absolute and relative impulse for each leg was calculated over this period.

## Statistical analyses

Data were placed into the Statistical Package for the Social Sciences (SPSS; IBM, New York, United States), Version 22. The data were screened for normality using

frequency tables, box-plots, histograms, z-scores and Shapiro–Wilk tests prior to hypotheses testing. One univariate outlier was detected and removed from three of the data sets, relative unilateral vertical GRF, relative and absolute upwards phase medio-lateral GRF. In order to satisfy normality, the medio-lateral GRF for the absolute upwards phase was transformed using the base 10-logarithm function. Following data screening, the final sample numbered 11–12 participants.

A 2×2 two-way ANOVA was used to evaluate the difference within peak resultant kettlebell force, absolute and relative GRF for: resultant, anterior–posterior, medio-lateral and vertical bilateral vectors for both the first and last 14 repetitions and the upwards and downwards phases. Additionally, absolute and relative unilateral GRF vectors were compared with a 2×2 two-way ANOVA between the ipsilateral and contralateral legs as well as the upwards and downwards phases. Temporal measures of kinetics were compared within different points of the kettlebell trajectory with two-tailed paired $t$-tests and a Bonferroni adjustment. An intra-repetition analysis compared the kinetics at six points of the kettlebell trajectory (highest point, midpoints, lowest points and end of the back swing), additionally peak bilateral GRF, maximum acceleration and peak resultant velocity were compared to their peak value (this was done to determine the different demands throughout a single repetition). The magnitude of the effect or effect size was assessed by Cohen's D (ESD) for $t$-tests and Cohen's F (ESF) for two-way ANOVA. Trials from both right and left hands were assessed. If the lifter performed an uneven number of repetitions with each hand, the side with the greatest number had repetitions randomly removed in order to allow for an even amount of pairs. Removed repetitions were evenly allocated between each minute. Within each minute, randomly generated numbers corresponding to each were used to determine removed repetitions. The magnitude of the paired $t$-test effect was considered trivial ESD < 0.20, small ESD 0.20–0.59, moderate ESD 0.60–1.19, large ESD 1.20–1.99, very large ESD 2.0–3.99 and extremely large ESD ≥ 4.0 (*Hopkins, 2010*). Statistical significance for the paired $t$-tests required $p < 0.001$. The magnitude of difference for the two-way ANOVA was reported as trivial ESF < 0.10, small ESF 0.10–0.24, medium ESF 0.25–0.39 and large ESF ≥ 0.40 (*Hopkins, 2003*). The two-way ANOVA required $p < 0.05$ for statistical significance.

## RESULTS

A total number of 972 repetitions were analysed for the 12 amateur kettlebell sport lifters, each performing an average of 13.9 ± 3.3 repetitions per minute. Grip strength of the hand that performed the last 3 min of the set had a reduction ($p = 0.001$, ESD = 0.77) of 9.8 ± 4.4 kg compared to pre-test results. Tables 1 and 2 show descriptive statistics for the three-dimensional GRF and kettlebell force during the first and last 14 repetitions for the absolute and relative values, respectively. The absolute peak resultant kettlebell force was significantly larger for the first repetition period compared to the last (i.e. first 14 vs last 14) when a full repetition was analysed (i.e. upwards and downwards phases combined) ($F_{(1.11)} = 7.42$, $p = 0.02$, ESF = 0.45).

Tables 3 and 4 show the descriptive statistics for the absolute and relative GRF of the ipsilateral and contralateral leg. At the point of peak resultant GRF for either the

**Table 1 Absolute mean (SD) resultant and three-dimensional GRF for the first and last 14 repetitions.**

|  | First 14 repetitions | | Last 14 repetitions | |
|---|---|---|---|---|
|  | Downwards | Upwards | Downwards | Upwards |
| GRF (N) | 1,766 (240) | 1,775 (277) | 1,782 (249) | 1,797 (285) |
| GRF x (N) | 47 (43) | 70 (33) | 59 (51) | 63 (42) |
| GRF y (N) | 308 (74) | 299 (80) | 320 (88) | 315 (92) |
| GRF z (N) | 1,736 (235) | 1,746 (271) | 1,748 (246) | 1,766 (278) |
| Resultant peak kettlebell force (N) | 809 (74) | 895 (76) | 826 (85) | 879 (101) |

Note:
x, medio-lateral, y, anterior–posterior, z, vertical.

**Table 2 Mean (SD) resultant and three-dimensional relative GRF (normalised to body weight (N)) for the first and last 14 repetitions.**

|  | First 14 repetitions | | Last 14 repetitions | |
|---|---|---|---|---|
|  | Downwards | Upwards | Downwards | Upwards |
| GRF (BW) | 2.06 (0.24) | 2.08 (0.31) | 2.08 (0.24) | 2.10 (0.31) |
| GRF x (BW) | 0.06 (0.05) | 0.08 (0.04) | 0.07 (0.06) | 0.07 (0.05) |
| GRF y (BW) | 0.36 (0.08) | 0.35 (0.10) | 0.37 (0.10) | 0.37 (0.11) |
| GRF z (BW) | 2.03 (0.24) | 2.04 (0.30) | 2.04 (0.25) | 2.07 (0.30) |

Note:
BW, weight body; x, medio-lateral; y, anterior–posterior; z, vertical.

**Table 3 Mean (SD) three-dimensional forces comparison of ipsilateral and contralateral with values shown as absolute values.**

|  | Ipsilateral downwards | Contralateral downwards | Difference | Ipsilateral upwards | Contralateral upwards | Difference |
|---|---|---|---|---|---|---|
| GRF (N) | 897 (133) | 939 (175) | 42 (4.6%) | 936 (110) | 949 (110) | 13 (1.38%) |
| GRF x (N) | 34 (16) | 59 (56) | 25 (53.7%) | 46 (25) | 33 (33) | 13 (32.9%) |
| GRF y (N) | 165 (42) | 154 (38) | 11 (6.9%) | 164 (39) | 146 (42) | 18 (11.6%) |
| GRF z (N) | 885 (126) | 939 (166) | 54 (5.9%) | 905 (93) | 942 (106) | 37 (4.0%) |
| Resultant impulse (N·s) | 380 (29) | 365 (64) | 15 (4.0%) | 382 (52) | 378 (63) | 4 (1.0%) |

Note:
x, medio-lateral; y, anterior–posterior; z, vertical.

ipsilateral and contralateral side, a large significant increase was found within the ipsilateral leg in the anterior–posterior vector ($F(1.11) = 885.15$, $p < 0.0001$, ESF = 7.00). In contrast, a large significant increase was found within the contralateral leg of the medio-lateral force vector over a full repetition for both the absolute GRF ($F(1.11) = 5.31$, $p = 0.042$, ESF = 0.67) and relative GRF ($F(1.10) = 9.31$, $p = 0.01$, ESF = 0.54). No significant differences were found for the absolute and relative impulse of the upwards or downwards phase. Figure 2 demonstrates a typical three-dimensional GRF of the ipsilateral and contralateral side.

**Table 4 Mean (SD) three-dimensional forces comparison of relative GRF (normalised to body weight N) ipsilateral and contralateral legs.**

| | Ipsilateral downwards | Contralateral downwards | Difference | Ipsilateral upwards | Contralateral upwards | Difference |
|---|---|---|---|---|---|---|
| GRF (BW) | 1.07 (0.14) | 1.11 (0.15) | 0.04 (3.7%) | 1.13 (0.14) | 1.11 (0.13) | 0.02 (1.8%) |
| GRF $x$ (BW) | 0.04 (0.02) | 0.08 (0.04) | 0.04 (66.7%) | 0.06 (0.04) | 0.04 (0.04) | 0.02 (40.0%) |
| GRF $y$ (BW) | 0.20 (0.05) | 0.18 (0.04) | 0.02 (10.5%) | 0.20 (0.06) | 0.16 (0.03) | 0.04 (22.2%) |
| GRF $z$ (BW) | 1.04 (0.13) | 1.07 (0.13) | 0.03 (2.8%) | 1.08 (0.19) | 1.08 (0.12) | 0 (0%) |
| Resultant impulse (BW·s) | 0.42 (0.50) | 0.44 (0.05) | 0.02 (4.7%) | 0.45 (0.05) | 0.43 (0.05) | 0.02 (4.6%) |

Note:
 BW, body weight; $x$, medio-lateral; $y$, anterior–posterior; $z$, vertical.

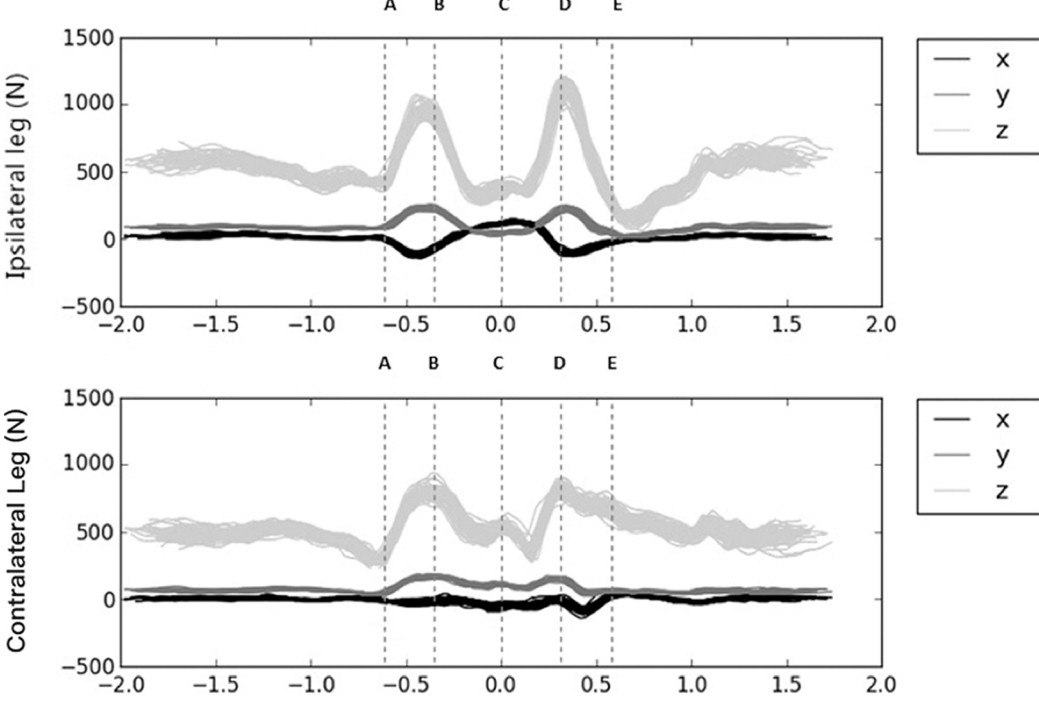

**Figure 2 Typical three-dimensional GRF of the ipsilateral and contralateral legs for an 87 kg athlete.** (A) midpoint (down), (B) lowest point (down), (C) end of back swing, (D) lowest point (up), (E) midpoint (up), $x$, medio-lateral; $y$, anterior–posterior; $z$, vertical.

Tables 5 and 6 provide data on how the kinematics and kinetics of the kettlebell snatch changed throughout the range of motion. Specifically, these tables list the relative times, resultant velocity and temporal changes in both kettlebell force and GRF with a comparison to their respective peak values during the downwards and upwards phases, respectively. Within the downwards phase there was no significant difference between peak bilateral GRF and bilateral GRF at the point of maximum acceleration, peak resultant velocity and resultant velocity at the midpoint. All other points had significant differences (see Tables 5 and 6).

Table 5 Mean (SD) temporal measures of resultant kettlebell force, resultant velocity and resultant GRF of the downwards phase.

| | Relative time (s) | Resultant kettlebell force (N) | Resultant velocity (m/s) | Resultant bilateral GRF (N) |
|---|---|---|---|---|
| Highest point overhead | −1.72 (0.49) | 222 (15)[1,5] | 0.28 (0.22)[1,5] | 1,054 (93)[1,4] |
| Midpoint | −0.60 (0.04) | 284 (53)[1,5] | 3.62 (0.21)[1,2] | 866 (153)[1,5] |
| Peak resultant velocity | −0.53 (0.05) | 466 (69)[1,5] | 3.81 (0.21) | 1,139 (165)[1,4] |
| Maximum acceleration | −0.40 (0.04) | 814 (75) | 3.23 (0.27)[1,4] | 1,660 (299) |
| Peak resultant GRF | −0.34 (0.11) | 775 (73) | 3.08 (0.29) | 1746.68 (217) |
| Lowest point | −0.31 (0.04) | 694 (79)[1,3] | 2.69 (0.34)[1,5] | 1,595 (276)[1,2] |
| End of the back swing | 0.00 (0.00) | 127 (43)[1,5] | 0.21 (0.08)[1,5] | 940 (169)[1,5] |

Notes:
The effect was trivial unless otherwise stated.
[1] Significantly ($p < 0.0001$) < peak value.
[2] Moderate ESD (0.6–1.2).
[3] Large ESD (1.2–2.00).
[4] Very large ESD (2.0–4.0).
[5] Extremely large ESD (>4.00).

Table 6 Mean (SD) temporal measures of resultant kettlebell force, resultant velocity and resultant GRF during the upwards phase.

| $n = 972$ | Relative time (s) | Resultant kettlebell force (N) | Resultant velocity (m/s) | Resultant bilateral GRF (N) |
|---|---|---|---|---|
| End of the back swing | 0.00 (0.00) | 127 (43)[1,6] | 0.21 (0.08)[1,6] | 940 (169)[1,6] |
| Lowest point | 0.32 (0.05) | 788 (112)[1,3] | 2.90 (0.37)[1,6] | 1,701 (320)[1,2] |
| Peak resultant GRF | 0.33 (0.05) | 798 (81)[1,3] | 2.89 (0.52)[1,5] | 1,768 (242) |
| Maximum acceleration | 0.39 (0.04) | 885 (86) | 3.51 (0.29)[1,5] | 1,634 (289)[1,2] |
| Peak resultant velocity | 0.51 (0.05) | 596 (62)[1,5] | 4.16 (0.23) | 1,095 (164)[1,5] |
| Midpoint | 0.60 (0.04) | 314 (38)[1,6] | 3.82 (0.20)[1,4] | 838 (122)[1,6] |

Notes:
The effect was trivial unless otherwise stated.
[1] Significantly ($p < 0.0001$) < peak value.
[2] Small ESD (0.2–0.6).
[3] Moderate ESD (0.6–1.2).
[4] Large ESD (1.2–2.00).
[5] Very large ESD (2.0–4.0).
[6] Extremely large ESD (>4.00).

# DISCUSSION

Three-dimensional motion analysis was used in this study to document kettlebell snatch kinetics of trained kettlebell sport athletes over a 6-min period. The main finding of this study was that the bilateral GRF was similar from the first and the last 14 repetitions, however, there were large significant differences within the resultant kettlebell force of the first and last 14 repetitions. Large differences were found between the ipsilateral and contralateral leg GRF within the anterior–posterior and medio-lateral vectors. Over the course of a single repetition, large differences in kettlebell force and GRF were evident as the kettlebell moved from the end of the back swing, to the lowest point, midpoint and highest point in the upwards and downwards phases. There were large differences in the bilateral GRF and the kettlebell force across different parts of the range of motion.

The kettlebell swing has received more attention than the kettlebell snatch in the scientific literature, possibly due to the relative ease of teaching and learning of the swing compared to the snatch. The kettlebell swing has been found to be an effective exercise for improving jump ability (*Jay et al., 2013*; *Lake & Lauder, 2012a*, *2012b*; *Otto et al., 2012*), strength (*Beltz et al., 2013*; *Lake & Lauder, 2012a*, *2012b*; *Manocchia et al., 2010*; *Otto et al., 2012*) and aerobic fitness (*Beltz et al., 2013*; *Falatic et al., 2015*; *Farrar, Mayhew & Koch, 2010*; *Hulsey et al., 2012*; *Thomas et al., 2013*). Additionally, the kettlebell swing was suggested to be a useful exercise for improving sprinter performance as it has a higher ratio of horizontal to vertical GRF compared to squat variations (*Beardsley & Contreras, 2014*). Previous research involving the (one armed) kettlebell snatch found the bilateral mechanical demands were similar to that reported for the two handed kettlebell swing in several ways (*Lake, Hetzler & Lauder, 2014*). For example, both exercises have a net vertical impulse greater than the net horizontal impulse (*Lake, Hetzler & Lauder, 2014*). Further, the marked difference between peak vertical and anterior–posterior GRF of the kettlebell snatch within this study support this. There appears to be little difference in the magnitude of the vertical impulse of the two kettlebell exercises, however, the horizontal impulse appears larger for the swing (*Lake, Hetzler & Lauder, 2014*). It is acknowledged that the two-handed kettlebell swing may be a more accessible choice for lower body power and strength training than the kettlebell snatch. However, the unilateral nature of the kettlebell snatch results in a different three-dimensional kinetic profile and may provide greater rotational core stability demands than the two-handed kettlebell swing. Muscle activation of the contralateral upper erector spinae has been shown to be higher than the ipsilateral portion of this muscle group during the one-armed swing and the same side during the two-armed swing (*Andersen et al., 2016*). Further, results of the current study indicated that the kettlebell snatch produced large effect size differences in two of the GRF vectors between the two legs. This suggests that the rotational component imposed different unilateral and force vector demands upon the entire body. The peak resultant force of the ipsilateral leg was found to occur later than the contralateral leg, which has also been shown in the unilateral dumbbell snatch (*Lauder & Lake, 2008*). This would suggest that during whole body exercises, holding the implement in one hand will place somewhat different demands, albeit of a modest magnitude, on the lower body even when it is functioning bilaterally.

This study demonstrates that with training, experienced kettlebell athletes are able to sustain consistent GRF over a prolonged 6-min set of kettlebell snatch, even though the kettlebell force over different points of the trajectory exhibited marked differences within each repetition. Interestingly, the peak resultant kettlebell force of the first 14 repetitions was significantly greater than the last 14 repetitions, suggesting that the kettlebell athletes were becoming fatigued at the end of the 6 min. This may be explained by the reduced handgrip strength that we observed, which anecdotally may be a limiting factor within kettlebell snatch competitions. The kettlebell athlete may attempt to take advantage of the less demanding phases of the kettlebell snatch to rest their grip, so as to prolong their performance.

Within different phases of the kettlebell snatch, there were marked differences in the intra-repetition kinetics. The differences in the kettlebell force throughout the range of motion may be an indicator of an efficient technique, thereby enabling prolonged performance of the kettlebell snatch. Peak resultant acceleration (in the upwards phase) occurred slightly after the lowest point of the trajectory, approximately after the kettlebell passed the knees. At the midpoint of the trajectory, the GRF of the upwards ($838 \pm 122$ N) and the downwards phases ($866 \pm 153$ N) was similar in magnitude to the body mass of the subjects ($860 \pm 113$ N). The low GRF in the overhead position would suggest that the bulk of the lower body's workload takes place as the kettlebell moves from the midpoint to the end of the back swing and back to the midpoint of the kettlebell snatch. The midpoint of the snatch is similar to a swing end point, as the swing follows the same trajectory and is analogous to the barbell snatch pull within weightlifting. Interestingly, the end of the back swing for the kettlebell snatch has the lowest kettlebell force of $121 \pm 45$ N, which is approximately half the weight force (235 N) of the 24 kg kettlebells. It has been suggested that this is one of two points (along with the overhead fixation position) of relative relaxation in the kettlebell snatch (*McGill & Marshall, 2012*). In fixation, the arm is positioned overhead with the kettlebell resting on the back of the distal forearm, with the handle sitting diagonally across the palm. This position has been shown to exhibit low variability in elite kettlebell sport lifters (*Ross et al., 2015*). This low variability may promote metabolic efficiency and safety by reducing the muscular effort required to hold the kettlebell overhead, additionally it is necessary to perform a valid repetition within kettlebell sport. Comparatively, this may not be applicable to hardstyle kettlebell snatch technique, as this style has a focus on effectiveness, rather than efficiency and does not generally involve determining valid repetitions. Following the point of relaxation at the end of the back swing, the forward swing transitions the kettlebell past the knees where the acceleration pull occurs. The acceleration pull is the most explosive movement of the kettlebell snatch and serves a similar function to the second pull in weightlifting. Resultant maximum acceleration occurred slightly after the lowest point suggesting it starts as the kettlebell passes the knees during the forwards swing of the snatch. Peak barbell velocity marks the end of the second pull phase within the barbell snatch (*Ho et al., 2014*), which suggests that the point of peak resultant velocity marks the end of the acceleration pull phase. Peak resultant velocity occurs just before the midpoint of the upwards phase. The kettlebells backwards and forwards swing in the snatch is somewhat similar to the first pull and transition phase in the weightlifting pull. As the kettlebell swings forward, it is progressively accelerated, until peak acceleration when the body of the lifter is in a more advantageous position. By having peak acceleration as the kettlebell passes the knees, force may be applied more efficiently, much like the power position in the weightlifting pull (*Newton, 2002*). The changes in the force applied to the kettlebell during its trajectory have been found to occur in conjunction with sequential muscular contraction and relaxation cycles (*McGill & Marshall, 2012*). In addition to these rapid contraction–relaxation cycles, kettlebell sport athletes use the lockout or fixation position to briefly rest between repetitions. Controlling the kettlebell overhead allows a valid repetition,

but it will also enable the athlete to regulate their pace, with longer and shorter pauses facilitating a slower or faster pace, respectively.

## CONCLUSION

In summary, the GRF and force applied to the kettlebell changes during different stages of the kettlebell snatch. Additionally, the kettlebell snatch places different external demands upon the ipsilateral and contralateral legs within the AP and ML force vectors. Thus, despite the kettlebell snatch being performed with two legs, each leg may be loaded differently, thereby offering a different stimulus to each leg. There are rapid changes within the kinetics during different phases of the lift. In the upwards and downwards phases there were extremely large significant intra-repetition differences within GRF, kettlebell velocity and force applied to the kettlebell. Applied force to the kettlebell during the first and last 14 repetitions at the point of peak resultant kettlebell force is altered over the course of a prolonged set, possibly due to muscular fatigue, which is further supported by a marked reduction in hand grip strength. The data from this investigation suggest that the kettlebell snatch may provide a unique training stimulus, compared to other exercises (e.g. barbell snatch), as it has a downwards phase and places different demands upon the ipsilateral and contralateral legs. In addition, within their respective sports the barbell and kettlebell snatches sit on different ends of the strength–endurance continuum.

## ACKNOWLEDGEMENTS

The authors would like to thank Angus McCowan for his assistance in the data analysis.

### Funding
The authors received no funding for this work.

### Competing Interests
Justin W.L. Keogh is an Academic Editor for PeerJ.

### Author Contributions
- James A. Ross conceived and designed the experiments, performed the experiments, analysed the data, contributed reagents/materials/analysis tools, wrote the paper, prepared figures and/or tables and reviewed drafts of the paper.
- Justin W.L. Keogh conceived and designed the experiments, wrote the paper and reviewed drafts of the paper.
- Cameron J. Wilson conceived and designed the experiments, wrote the paper and reviewed drafts of the paper.
- Christian Lorenzen conceived and designed the experiments, analysed the data, contributed reagents/materials/analysis tools, wrote the paper and reviewed drafts of the paper.

## Human Ethics

The following information was supplied relating to ethical approvals (i.e. approving body and any reference numbers):

The Australian Catholic University's ethics review panel granted approval for this study to take place (ethics number 2012 21V).

## Data Availability

The raw data has been supplied as Supplemental Dataset Files.

## Supplemental Information

Supplemental information for this article can be found online at http://dx.doi.org/10.7717/peerj.3111#supplemental-information.

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
