# Peer review of "External kinetics of the kettlebell snatch in amateur lifters"

_PeerJ, doi:10.7717/peerj.3111_

## Round 0.1 · original submission · Minor Revisions

As you can see, we have two excellent reviews to help you to improve the quality of your manuscript. Please, carefully follow the suggestions of both reviewers

·

Basic reporting

With regards to the main points of basic reporting, the authors have done an excellent job. With the exception of two areas that I feel could be clearer all of the main points for this section have been thoroughly addressed. The two main points are as follows:

1) Athlete kettlebell ‘background’, pre-study technique, study technique
a. Please clarify whether the subjects were competitive GS athletes and whether their minimum of 3 years of kettlebell training was in GS. If not…
b. …were the 6 training sessions that they completed before data collection based on GS snatch technique or their ‘typical’ snatch technique, whatever that may have been?
c. I think it might be worth noting that there are some considerable differences between GS and Hardstyle snatch technique, the main one being that GS technique is based on trying to maximise mechanical efficiency, whereas in some ways Hardstyle technique goes out of its way to maximise the mechanical demand by emphasising the hip hinge.

2) Dependent variable clarity
a. I think there needs to be a bit more clarity in both the way you describe your dependent variables and the way you present them. For example, you describe calculating resultant values, but provide no rationale for this. Lake and Lauder (2012) used a similar approach, but they did so to simplify their comparison of the mechanical demands of two-handed swing exercise to those of back squat and jump squat exercise. In your case I think it might be more effective to focus on horizontal (anterior-posterior) and vertical data as this will enable you to quantify elements of technique and the effect that performance duration has on them.

b. The other point that I think could benefit from greater clarity is the way you refer to GRF and ‘applied’ force. I think you may be better served by briefly explaining that: ‘…GRF provides a measure of the force applied to the lifter and kettlebell system centre of mass, which is of interest and ultimately defines absolute external mechanical demand, but provides no insight in to the kinetics of the kettlebell. By recording the acceleration of the kettlebell and multiplying this by its mass kettlebell force can be used to consider this with greater clarify…’. You could refer to these thereafter as something like ‘GRF and kettlebell force’ or ‘system and kettlebell force’

c. Finally, I think you make more of the way you present that unilateral force data; maybe consider the typical side-to-side differences at least to demonstrate why using this approach might be useful.

I have included some minor points that should be addressed in the annotated version of your manuscript that I have attached.

Experimental design

The experimental design is robust and enables the authors to answer their research question, which is both relevant and meaningful; the results of this study will help inform applied sports scientists and strength and conditioning practitioners.
Other than the way some of the dependent variables have been defined/described (please see my discussion of this in the first box), all of the main points have been addressed very clearly and thoroughly.
I have included some minor points that should be addressed in the annotated version of your manuscript that I have attached.

Validity of the findings

The authors have done an excellent job in addressing the main points of this section. However, it could be argued that some of the points I have raised about the clarity of the dependent variables may compromise this slightly, but they will be easily able to rectify this.
I have included some minor points that should be addressed in the annotated version of your manuscript that I have attached.

Additional comments

Thank you for giving me the opportunity to review this manuscript. The authors set out to expand understanding of the biomechanics of the kettlebell snatch by studying both 3D kinematics and kinetics during the first and last minute of a 6 minute bout of kettlebell snatch exercise; uniquely, they also considered ground reaction forces independently recorded from the lifting and non-lifting side.
The authors are to be commended for applying such a well-designed experimental approach to their research question and for the quality of their writing. They have provided a very well written manuscript that provides new and potentially very useful insights for applied sports scientists and strength and conditioning practitioners.
The final point that I would like to make is that they are to be commended for demonstrating their enthusiasm for the subject through their writing -- thank you!

·

Basic reporting

1.1. There could be an image depicting the kettlebell snatch phases.
1.2. There are some punctuation problems:
• “..., however their traditional uses…” (background, line 4, missing comma after however).
• “…kettlebell exercises, however the horizontal impulse…” (line 240, missing comma after however).
• “Kettlebell sport, also referred to as Girevoy Sport (GS)…” (line 18, missing comma after GS).
• “Prior to taking part in the study the participants performed…” (lines 90-91, missing comma after study).
• “…on the lower body even when it’s functioning bilaterally.” (line 253. You should not use contractions/short forms in academic papers. Suggested answer: “… it is functioning bilaterally.”).

1.3. Grammatical problem (word choice and unnecessary words):

• “…in the hand by which they hold the kettlebell during this ten minute period.” (line 30, suggested option: “…in the hand they hold the kettlebell with….”).

• “Large effect size differences in the GRF were found between the ipsilateral and contralateral legs…” (line 223, too wordy and confusing. Suggested option: “Large differences in the GRF….”).
• “a prolonged six-minute set of the kettlebell snatch…” (lines 256-257, unnecessary word: the. Suggested answer: “a prolonged six-minute set of kettlebell snatch…).
• “The differences in the applied force throughout the range of motion may be indicative of an efficient technique,” (lines 267-268, wrong word: indicative. Suggested answer: “…may be an indicator of an efficient technique,”.
• “In addition, the kettlebell snatch places…” (line 304. It could be advisable to change the connector “in addition” for “additionally” in order to avoid word repetition).
• “During the upwards phase and downwards phases there were extremely large…” (line 308. It could be advisable to change the connector “during” for “in” to avoid word repetition).

Experimental design

2. Experimental Design

2.1. In GS male professional athletes compete with 32kgs, whereas amateurs compete with 24kgs. It would be advisable to refer to your sample as amateur kettlebell lifters or amateur athletes.
2.2 In relation to the previous point, the sample’s documented training and competition pace also reinforces the idea of being amateur athletes in kettlebell lifting.
2.3. It would be advisable to consider kettlebell lifting as strength-endurance instead of resistance training (line 20).
2.4 In line 314, you compare kettlebell snatch and barbell snatch. Whilst they might be technically similar, their physiological demands are extremely different as to be compared in this context. It could be perceived as comparing 1 maximum repetition to over 200 submaximal repetitions.
2.5 In Kettlebell Sport, there could be significant technical differences among lighter and heavier athletes. This should be important for your study considering the range in weight categories among your sample.
2.6 The overhead fixation in GS with proper technique is supposed to be on the distal end of the forearm instead of the wrist (line 281).
2.7. What do you refer to by “metabolic efficiency” in line 283? Please, elaborate this point.
2.8. Instead of talking about scoring points, it would be better to talk about valid repetitions. This problem repeats along the paper.

Validity of the findings

3. Validity of the Findings

3.1 There is no reference referring to the link between grip strength and endurance for kettlebell snatch. It is also not presented as a speculation but as a fact, so it should be fixed.

Additional comments

4. General comments

4.1 The topic of your study is quite innovative and increasingly meaningful considering the raising popularity of GS worldwide.
4.2 It would be advisable to use proper kettlebell sport terminology like amateur to refer to athletes competing with 24kg kettlebells and the word repetitions instead of points.
4.3. Basing on 2.1. and 2.2., It is advisable to change the title of this paper to:
• Option 1: External kinetics of the kettlebell snatch in amateur athletes.
• Option 2: External kinetics of the kettlebell snatch in amateur lifters.

---

## Round 0.2 · accepted · Accept

Thank you very much for considering the suggestions of the reviewer. I think that your manuscript is ready for publication.

·

Basic reporting

The authors have achieved all of the main targets of this section very clearly and thoroughly.

Experimental design

The authors have achieved all of the main targets of this section very clearly and thoroughly.

Validity of the findings

The authors have achieved all of the main targets of this section very clearly and thoroughly.

Additional comments

The authors should be commended for doing a really excellent job with their revisions - thank you!
They have addressed all of my comments/queries thoroughly in addition to making some excellent additions.

·

Basic reporting

All the requested changes have been added in the reviewed version of this paper in terms of language (grammar and punctuation), terminology and multimodal resources (images).

Experimental design

The conceptual changes regarding the nature of GS have been successfully applied by the author.

Validity of the findings

The speculation regarding grip strength and endurance for kettlebell snatch competitions has been corrected and identified as such.

Additional comments

All the suggested changes have been made. The reviewer thanks the author for his commitment and good will.